# Numerical investigation of LDL nanoparticle collision in coronary artery grafts with porous wall and different implantation angles and two state of inlet velocity

**Reza Karimian** *, Mohsen Saghafian, Ebrahim Shirani

Department of Mechanical Engineering, Isfahan University of Technology, Isfahan, Iran

* r.karimian@alumin.iut.ac.ir

**Data Availability Statement:** All relevant data are within the manuscript and its Supporting Information files. And in the stable, public

## Abstract

This study aimed to reduce the risk of graft occlusion by evaluating the two-phase flow of blood and LDL nanoparticles in coronary artery grafts. The study considered blood as an incompressible Newtonian fluid, with the addition of LDL nanoparticles, and the artery wall as a porous medium. Two scenarios were compared, with constant inlet velocity (CIV) and other with pulsatile inlet velocity (PIV), with LDL nanoparticles experiencing drag, wall-induced lift, and induced Saffman lift forces, or drag force only. The study also evaluated the concentration polarization of LDLs (CP of LDLs) near the walls, by considering the artery wall with and without permeation. To model LDL nanoparticles, the study randomly injected 100, 500, and 1000 nanoparticles in three release states at each time step, using different geometries. Numerical simulations were performed using COMSOL software, and the results were presented as relative collision of nanoparticles to the walls in tables, diagrams, and shear stress contours. The study found that a graft implantation angle of 15˚ had the most desirable conditions compared to larger angles, in terms of nanoparticle collision with surfaces and occlusion. The nanoparticle release modes behaved similarly in terms of collision with the surfaces. A difference was observed between CIV and PIV. Saffman lift and wall-induced lift forces having no effect, possibly due to the assumption of a porous artery wall and perpendicular outlet flow. In case of permeable artery walls, relative collision of particles with the graft wall was larger, suggesting the effect of CP of LDLs.

## 1.Introduction

In 2015, coronary artery disease was diagnosed in 110 million people worldwide, resulting in 8.9 million deaths, making it the leading cause of death, accounting for 15.6% of all annual deaths [1]. The conventional treatments for coronary artery disease include percutaneous coronary intervention (PCI) or coronary artery bypass graft (CABG). However, it remains uncertain whether these treatments can effectively reduce the risk of heart attacks, extend patients' lives, or improve their condition [2]. CABG involves surgically restoring blood flow to a

repository and the relevant link is as below: https://github.com/Rkarimian97/paper-original-files.git.

**Funding:** The authors received no specific funding for this work.

**Competing interests:** The authors have declared that no competing interests exist.

**Abbreviations:** D, Channel diameter (artery) (m); $d_p$, Nanoparticle diameter (m); E, Porosity coefficient; g, Gravitational acceleration '$(m.s^{-2})$; K, Permeability of porous media $(cm^2)$; $m_p$, Nanoparticle mass (kg); $\vec{n}$, Unit vector; P, Fluid pressure field (Pa); $r_p$, Nanoparticle radius (m); U, Nanoparticle velocity field $(m.s^{-1})$; $u_{||}$, Velocity parallel to the surface $(m.s^{-1})$; V, Fluid velocity field $(m.s^{-1})$; Θ, Graft implantation angle (°); μ, Fluid viscosity (Pa.s); ρ, Fluid density $(kg.m^{-3})$; $ρ_p$, Nanoparticle density $(kg.m^{-3})$; τ, Wall shear stress $(N.m^{-2})$.

blocked coronary artery in cases with 50–90% stenosis, a condition characterized by the thickening and inflexibility of the artery wall, leading to narrowed blood flow. Stenosis occurs due to the accumulation of cholesterol, lipids, and cellular residuals on the coronary artery wall, which obstruct normal blood flow [3]. Data suggest that within one year of coronary artery bypass grafting, 10–15% of vein grafts experience stenosis, and after 6 to 12 years, more than 70% of vein grafts become occluded [4].

Stenosis is linked to the concentration of lipids near the artery wall. This hypothesis suggests that a higher concentration of lipids at the blood/artery boundary can affect the artery wall's permeability [1]. Fan et al. investigated the impact of the concentration polarization of LDLs (CP of LDLs) and flow hemodynamics on grafts, examining implantation angles of 30, 45, and 60˚ and S-type geometry. Their findings revealed that the best flow hemodynamics occurred at a graft implantation angle of 30˚, with an increase in angle leading to worsened hemodynamic parameters. Additionally, the CP of LDLs was at its optimum at the 30˚ implantation angle [2] which is considered the lowest implantation angle. Using numerical simulations, Kabinejadian et al. explored how the graft's shape can affect the flow hemodynamics and stenosis. Their results showed that the rotational flow formation can increase shear stress on the wall, shorten the remaining time of the particle in the anastomotic region and coronary artery, and promote the duration of artery openness [3]. Fallahi et al. analyzed the hemodynamics of bypass grafts implanted on coronary arteries with elastic walls and varying stenosis levels. They considered blood as a Newtonian fluid with unsteady flow and found that pulsatile flow produced more accurate data. Moreover, the velocity field and shear stress on the wall were lower compared to steady-state flow [4].

Vishesh Kashyap and colleagues studied the curves in coronary arteries. They measured the bend angle, ranging from 0 to 60 degrees. Their findings highlighted that the arteries' curves impact how blood flows. Generally, higher curvature angles tend to increase the risk of blockages [5].

S. Kenjereš, J.P. van der Krieke, and C. Li conducted a study investigating blood flow and the transport of low-density lipoprotein (LDL) mass in an axisymmetric geometry resembling a stenosis in a diseased coronary artery. Their analysis involved considering the arterial wall as a porous medium comprising multi-layered structures of varying thickness. The primary objective was to develop a mathematical model based on an anatomically realistic representation of the internal structure of the arterial wall, aiming to establish a robust understanding of LDL transport through endothelial cells [6].

Md Foysal Rabbi et al. studied the impact of geometric variations on intravascular and near-wall hemodynamics. Using four patient-specific geometries—featuring branches of 70˚, 95˚, and 135˚, alongside three 90˚ branches—their findings highlighted the significant influence of geometry, particularly branch angles, on left coronary artery hemodynamics and the risk of occlusion. Notably, they observed substantial effects on near-wall hemodynamics due to varying branch angles [7].

Giuseppe De Nisco et al. explored incorporating blood rheological properties into computational fluid dynamics (CFD) simulations of coronary hemodynamics. They analyzed the right coronary artery (RCA) geometry in 144 patients with varying stenosis levels, considering two models: (i) a Newtonian and (ii) a non-Newtonian blood rheology model. Their findings indicate that blood rheology minimally impacts wall shear stress (WSS) and helical flow profiles. Hence, the use of a Newtonian assumption for blood rheology seems appropriate in coronary hemodynamic simulations [8].

This research aims to identify high-risk areas for re-occlusion in coronary artery grafts implanted at different angles. The study was conducted using two conditions: constant and pulsatile inlet velocities. Permeable and non-permeable walls were used to evaluate fluid outlet

and the CP of LDLs. To investigate the impact of nanoparticle numbers on their collision with the graft wall, three release modes were evaluated. Nanoparticles were studied under two force application conditions. In section 2, the problem definition is followed by the presentation of geometry and equations. The grid study and validation of the results are presented in section 3. In section 4 the results are presented and analyzed. Finally, the conclusion is written in section 5.

## 2. Problem description

Solid spherical LDL nanoparticles were introduced randomly into the flow and analyzed under two conditions: a) considering only the drag force, and b) accounting for the drag, wall-induced lift, and Saffman lift forces. The arterial wall, modeled as a porous medium, was studied in both permeable and non-permeable states to investigate concentration polarization of LDLs (CP of LDLs) near the walls. In the permeable wall scenario, a vertical outflow was simulated at the external vessel wall. The study focused on analyzing two distinct flow regimes: constant and pulsatile. Due to the nanoparticles' diluted concentration and very low load factor, the impact of the discrete phase on the continuum phase was disregarded [9]. The problem was addressed using a one-way method, initially solving the flow analysis equations and subsequently employing the obtained results to solve the equations concerning nanoparticles.

### 2.1. Geometry, governing equations and boundary conditions

Fig 1A illustrates the geometry of the problem, with the variable parameter being the θ angle (i.e., the angle of the implanted graft in relation to the main vessel). The θ angles selected were 15, 22.5, 30, 37.5, and 45°. The total vessel length was 100 mm, and both the main artery and graft diameter were 4 mm [10]. The main channel had a diameter of 3.3 mm [11], and the artery wall thickness (porous medium) was 0.35 mm [12]. COMSOL Multiphysics, utilizing the finite element method, was employed for simulating the system dynamics. The simulations were conducted over three cardiac cycles.

This study focuses on a single-phase, incompressible fluid with Newtonian[14,15], properties similar to blood, with density of 1060 $kg.m^{-3}$ and dynamic viscosity of 0.004 $Pa.s$ [15,16]. The continuity and Navier-Stocks equations are expressed as [17]:

$$\nabla.V = 0 \tag{1}$$

$$\rho\left(\frac{DV}{Dt}\right) = \rho g - \nabla p + \mu\nabla^2 V \tag{2}$$

Here $V$, $\rho$, $p$, and $\mu$ respectively are the velocity, density, pressure, and viscosity of the fluid. The wall was assumed to be inflexible and had no-slip boundary conditions [18]. Two inlet velocity states were used: a constant inlet velocity (CIV) of 25.45 $cm.s^{-1}$, and a pulsatile inlet velocity (PIV) shown in Fig 1B [13]. The CIV problem was simulated as a steady-state case, while the PIV problem was simulated for three cycles.

The porous medium was simulated by Brinkman equation using Eq (3).

$$\frac{1}{\epsilon}\rho(\nabla.V)V\frac{1}{\epsilon} = \nabla.[-\rho I + K] - \left(\mu k^{-1} + \beta\rho(V) + \frac{Q_m}{\epsilon^2}\right)V \tag{3}$$

$$\beta = \frac{-1.75}{\sqrt{150*\epsilon}} \tag{4}$$

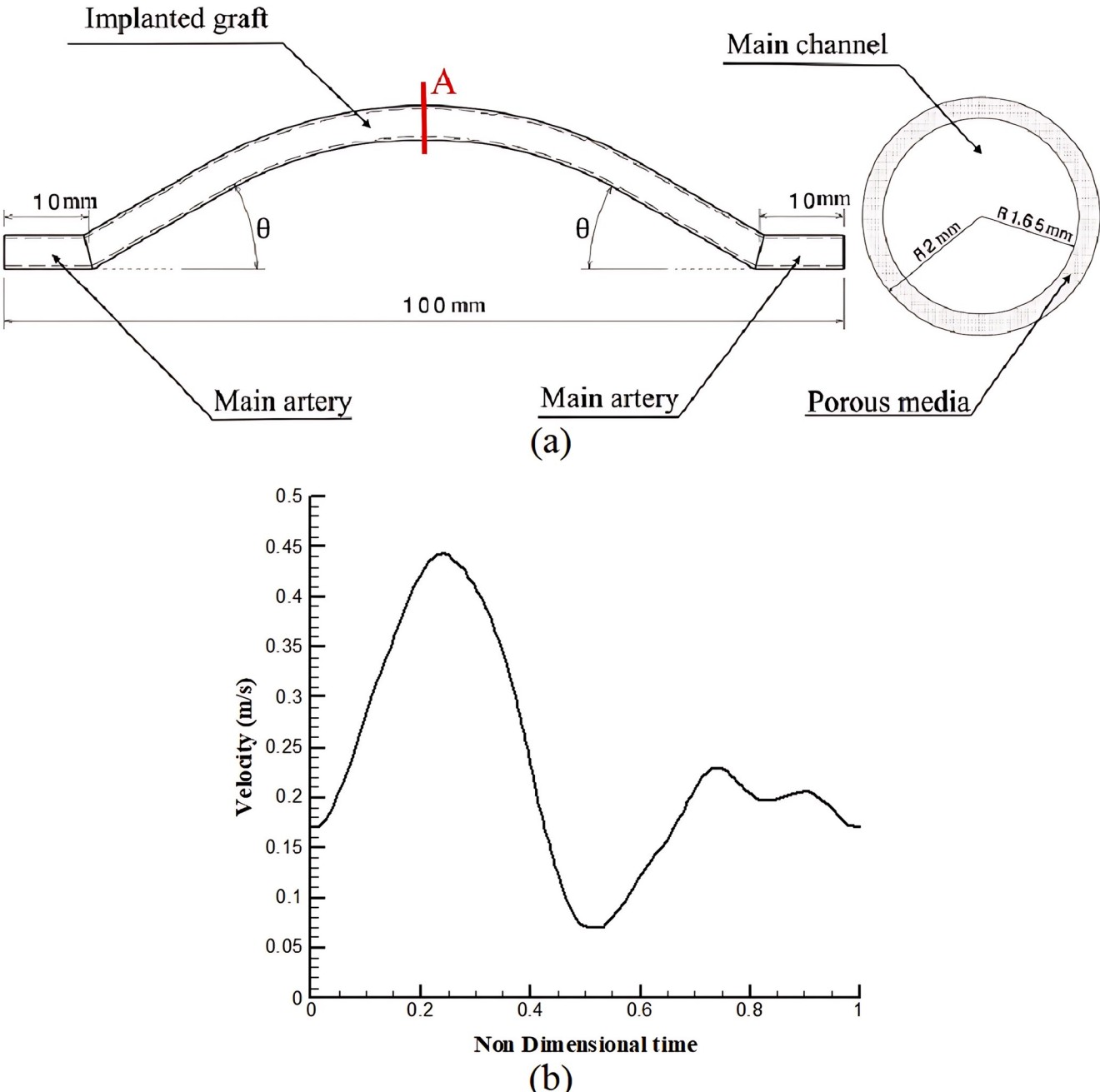

**Fig 1.** A) geometry B) Pulsatile velocity applied to the inlet [13].

$$Q_m = \rho \nabla.V \tag{5}$$

$$K = \frac{\mu}{\epsilon}\left(\nabla V + (\nabla V)^T\right) - \frac{2}{3}\frac{\mu}{\epsilon}(\nabla.V)I \tag{6}$$

Porosity $(\epsilon)$ and permeability $(k)$ values of the porous medium are provided [19]:

$\epsilon = 0.15$ and $k = 2*10^{-14} cm^2$

Wilnes et al. measured an outlet velocity from the artery wall of approximately $10^{-8} m.s^{-1}$. In the with-permeation case, the outer surface of the porous medium was assumed to have a normal outlet velocity of $10^{-8} m.s^{-1}$ [2,20,21]. LDL particles were modeled as spherical solid nanoparticles with a mean diameter of 23 nm and a density of 1040 $kg.m^{-3}$ [22]. Stick boundary conditions were applied to the walls. The impact of the nanoparticle population at the inlet was assessed by considering different numbers of nanoparticles (3000, 15000, and 30000) and injecting 100, 500, and 1000 nanoparticles randomly at the inlet in each 0.1-second time step. The solution time was three cycles, and the governing equation was the second law of Newton (Eq 7).

$$\frac{d(m_p \overrightarrow{V})}{dt} = \overrightarrow{F}_t \tag{7}$$

Here, $m_p$ represents nanoparticle mass, while $\overrightarrow{V}$ and $\overrightarrow{F}_t$ indicate velocity and superimposed forces fields, respectively. The forces driving nanoparticle motion in fluid consist of drag, Saffman inductive force, and wall-induced lift [23]. The drag force is expressed as follows:

$$\overrightarrow{F}_D = \frac{1}{\gamma_p} m_p (\overrightarrow{u} - \overrightarrow{v}) \tag{8}$$

Here, $m_p$ represents nanoparticle mass; $(\overrightarrow{u} - \overrightarrow{v})$ indicates the velocity disparity between the nanoparticle and fluid. And

$$\gamma_p = \frac{\rho_p d_p^2}{18\mu} \tag{9}$$

Here, $\rho\_p$, $d_p$, and μ denote the density, diameter, and viscosity of nanoparticles [24], respectively. The Saffman inductive force can be mathematically represented as [25]:

$$\overrightarrow{F}_s = 6.46 r_p^2 L_v \sqrt{\mu \frac{|u - v|}{|L_v|}} \tag{10}$$

Where,

$$L_v = (\overrightarrow{u} - \overrightarrow{v}) \times [\nabla \times (\overrightarrow{u} - \overrightarrow{v})] \tag{11}$$

Eq 12 represents the wall-induced lift force in the presence of a particle in fluid flow.

$$\overrightarrow{F}_L = \rho \frac{r_p^4}{D^2} \beta [\beta G_1(s) + \gamma G_2(s)] \overrightarrow{n} \tag{12}$$

In which, s shows the normalized distance to the first parallel wall. G is the location function of each particle at any time; $\overrightarrow{n}$ represents the unit vector from the nearest point in the first parallel wall; D stands for the channel diameter; while β and γ can be respectively determined by:

$$\beta = |D(n.\nabla)u_\parallel| \tag{13}$$

$$\gamma = |\frac{D^2}{2}(n.\nabla)^2 u_\parallel| \tag{14}$$

Here $u_\parallel$ indicates the velocity component parallel to the surface [25].

$$u_\parallel = (I - n \otimes n)u \tag{15}$$

## 3. Grid study and validation

### 3.1. Grid study

This study utilized a type-O grid structure with a finer grid near the wall regions for improved accuracy. Five different grids were used in the grid study for each graft implantation angle to determine variations in wall shear stress along the peripheral direction of A cross-section, as shown in Fig 1A. Table 1 lists the maximum difference in shear stress between each grid and the finest grid ($\Delta\tau_{max}$).

$$\Delta\tau_{max} = \text{Max}|\tau_w(\theta)_{\text{finest mesh}} - \tau_w(\theta)_{\text{other mesh}}|$$

Table 1 displays $\Delta\tau_{max}$ for different graft implantation angles and node numbers. Based on this table, 80000 nodes were chosen for each implantation angle based on the identified errors.

### 3.2. Validation

To validate the results, Sun et al.'s reports were used [26]. Their simulation considered the deposition of LDL nanoparticles in a laminar flow passing through an artery, including drag,

**Table 1. Maximum shear stress variation for different graft implantation angles and node numbers.**

| graft implant angle [°] | Number of mesh | $\Delta\tau_{max}$ [N.m$^{-2}$] | error percentage |
|---|---|---|---|
| 15 | 22500 | 0.14024 | 9.27% |
| | 38600 | 0.05458 | 3.5% |
| | 54000 | 0.02273 | 1.54% |
| | 80600 | 0.01048 | 0.71% |
| | 100500 (finest mesh) | 0 | 0 |
| 22.5 | 23000 | 0.16918 | 10.12% |
| | 40000 | 0.07792 | 4.67% |
| | 55000 | 0.03357 | 2.15% |
| | 82200 | 0.01296 | 0.83% |
| | 101200 (finest mesh) | 0 | 0 |
| 30 | 22300 | 0.14822 | 11.03% |
| | 38200 | 0.08483 | 5.78% |
| | 53200 | 0.03399 | 1.98% |
| | 80000 | 0.0115 | 0.67% |
| | 100200 (finest mesh) | 0 | 0 |
| 37.5 | 22500 | 0.19223 | 9.35% |
| | 38600 | 0.13152 | 7.963% |
| | 54000 | 0.06194 | 2.885% |
| | 80600 | 0.01009 | 0.47% |
| | 100500 (finest mesh) | 0 | 0 |
| 45 | 22500 | 0.30081 | 10.04% |
| | 38400 | 0.21483 | 7.1% |
| | 53700 | 0.09648 | 3.153% |
| | 81000 | 0.01561 | 0.51% |
| | 100150 (finest mesh) | 0 | 0 |

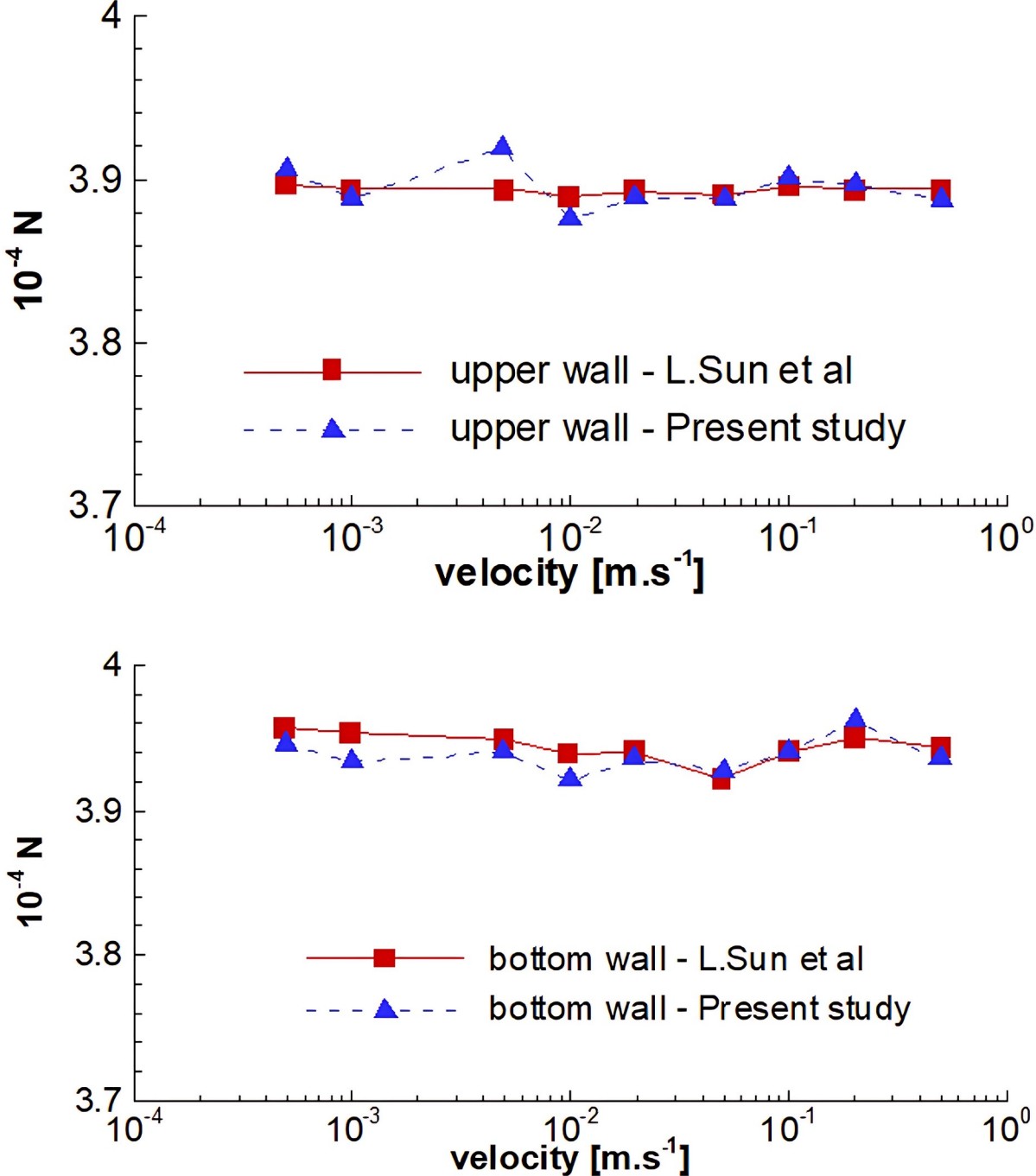

**Fig 2. Trapped particle count versus entrance velocities, upper, and lower walls.**

Brownian, gravity, Buoyancy, and Van der Waals forces. LDL was simulated as a solid nanoparticle with a density of 1060 kg.m$^{-3}$, and blood was simulated as a single-phase incompressible fluid with a density of 1030 kg.m$^{-3}$ and dynamic viscosity of 0.0035 Pa.s. Results included a plot of the number of deposited 20 nm diameter nanoparticles versus inlet velocity, ranging from 0.0005 m.s$^{-1}$ to 0.5 m.s$^{-1}$, with 100,000 nanoparticles included. Fig 2 displays the number

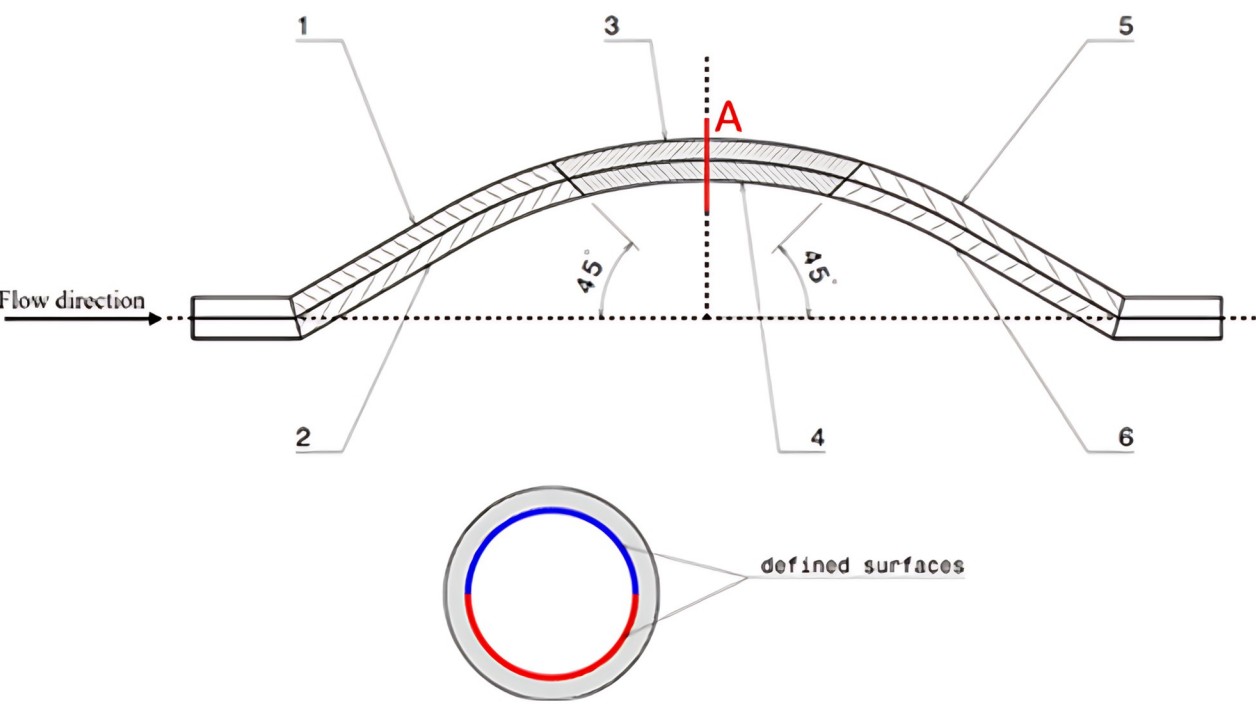

**Fig 3. Surfaces and section defined on the geometry.**

of nanoparticles trapped in upper and lower walls for various inlet velocities. The comparison between the results of the paper chosen for validation and the simulations in the present study indicates differences of less than 5% in the cases with the maximum discrepancies.

## 4. Results

The study aims to compare LDL nanoparticles colliding with graft walls at different implantation angles to identify the areas with the highest collision probability. The objective is to determine the ideal geometry by injecting 100–1000 nanoparticles randomly every 0.1-second at the inlet. A dimensionless parameter is used to determine collision distribution among surfaces. Surfaces exhibiting higher collision rates are prone to developing stenosis. In this study, specific surfaces were defined, as illustrated in Fig 3. The study considers two conditions: with/without wall permeation, various force applications, and two inlet velocity modes. The study also explores the wall shear stress, which plays a crucial role in stenosis.

### 4.1. Surfaces and parameter

Fig 3 shows six bent semi-cylindrical surfaces in the internal boundary of the porous medium. Surfaces 1 and 2 represent the graft inlet, surfaces 3 and 4 represent the middle parts of the graft, and surfaces 5 and 6 represent the graft outlet. The study was conducted in three-dimensional conditions. Results are presented using the dimensionless parameter NCP.

$$\text{Normalized Collied Particles} \overrightarrow{N}CP$$

$$= 10000 * \left( \frac{\text{number of nanoparticles colliding the surface}}{\textit{total number of injected nanoparticles}} \right)$$

NCP value may vary slightly in simulations due to random nanoparticle injection.

## 4.2. Comparison of results for permeation and non-permeation from the wall using two different modes of force application

Table 2 compares the effects of wall permeation on nanoparticle collision behavior under two different force application modes. For 500 nanoparticles released in each time step, the non-permeation state exhibited lower values of Normalized Collided Particles (NCP) compared to the permeation state. This was due to the normal flow strengthening the boundary between the fluid and porous medium, directing more nanoparticles towards the walls in the non-permeation state. Both states demonstrated a similar trend in nanoparticle collision behavior with the surfaces. The comparison highlighted the significance of Concentration Polarization of

**Table 2. Effect of implantation angle and surface type on NCP of Grafts.**

| Surfaces | graft implant angle [°] | | | | | | | | | |
|---|---|---|---|---|---|---|---|---|---|---|
| | 15 | | 22.5 | | 30 | | 37.5 | | 45 | |
| | CIV | PIV | CIV | PIV | CIV | PIV | CIV | PIV | CIV | PIV |
| non-permeation from the wall and injection of 500 nanoparticles at the inlet at each time step and applying all forces | | | | | | | | | | |
| 1 | 0 | 0 | 0 | 0 | 0 | 17 | 113 | 563 | 352 | 703 |
| 2 | 10 | 17 | 24 | 75 | 83 | 195 | 123 | 187 | 213 | 233 |
| 3 | 0 | 0 | 0 | 0 | 0 | 0 | 0 | 0 | 0 | 0 |
| 4 | 0 | 0 | 0 | 0 | 0 | 0 | 0 | 0 | 0 | 0 |
| 5 | 15 | 23 | 175 | 257 | 308 | 347 | 686 | 373 | 1376 | 1533 |
| 6 | 0 | 0 | 0 | 0 | 0 | 0 | 0 | 0 | 0 | 8 |
| permeation from the wall and injection of 500 nanoparticles at the inlet at each time step and applying all forces | | | | | | | | | | |
| 1 | 0 | 0 | 0 | 0 | 0 | 27 | 177 | 967 | 406 | 1059 |
| 2 | 12 | 64 | 33 | 87 | 144 | 236 | 155 | 273 | 321 | 340 |
| 3 | 0 | 0 | 0 | 0 | 0 | 0 | 0 | 0 | 0 | 0 |
| 4 | 0 | 0 | 0 | 0 | 0 | 0 | 0 | 0 | 0 | 0 |
| 5 | 34 | 37 | 266 | 309 | 401 | 427 | 1037 | 584 | 2447 | 1753 |
| 6 | 0 | 0 | 0 | 0 | 0 | 0 | 0 | 0 | 24 | 12 |
| applying all forces (A) and applying only drag force (D) and injection of 1000 nanoparticles at the inlet at each time step and pulsatile inlet velocity | | | | | | | | | | |
| | A | D | A | D | A | D | A | D | A | D |
| non-permeation from the wall | | | | | | | | | | |
| 1 | 0 | 0 | 0 | 0 | 26 | 26 | 630 | 620 | 756 | 753 |
| 2 | 40 | 40 | 52 | 52 | 191 | 187 | 202 | 202 | 365 | 365 |
| 3 | 0 | 0 | 0 | 0 | 0 | 0 | 0 | 0 | 0 | 0 |
| 4 | 0 | 0 | 0 | 0 | 0 | 0 | 0 | 0 | 0 | 0 |
| 5 | 48 | 48 | 171 | 166 | 357 | 357 | 703 | 703 | 1887 | 1887 |
| 6 | 0 | 0 | 0 | 0 | 0 | 0 | 0 | 0 | 10 | 10 |
| permeation from the wall | | | | | | | | | | |
| 1 | 0 | 0 | 0 | 0 | 34 | 34 | 1055 | 1055 | 1646 | 1646 |
| 2 | 50 | 50 | 77 | 77 | 257 | 257 | 286 | 286 | 421 | 421 |
| 3 | 0 | 0 | 0 | 0 | 0 | 0 | 0 | 0 | 0 | 0 |
| 4 | 0 | 0 | 0 | 0 | 0 | 0 | 0 | 0 | 0 | 0 |
| 5 | 58 | 58 | 251 | 251 | 437 | 437 | 942 | 942 | 2447 | 2447 |
| 6 | 0 | 0 | 0 | 0 | 0 | 0 | 0 | 0 | 21 | 21 |

Comparing permeable and non-permeable walls, and forces acting on nanoparticles (A) applying all forces and (D) just drag force.

LDLs (CP of LDLs) near the walls. This polarization increases the number of nanoparticles colliding with the surfaces, consequently heightening the risk of stenosis onset and intensification.

Table 2 also includes the NCP values for different graft implantation angles under two modes of force application, where 1000 nanoparticles were released in each time step. The non-permeation state showed a negligible difference in NCP in some cases, due to the normal output flow between the fluid-porous medium boundary, which reduced the impact of wall-induced lift and Saffman forces. In most cases, however, the results were the same. In the permeation state, stronger normal velocity in the fluid-porous medium boundary resulted in completely coinciding results, indicating no impact of Saffman and wall-induced lift forces. The simulations confirmed the results obtained for all cases, and Table 2 is presented as an example of the findings.

## 4.3. Results based on NCP

Based on Table 3 and Fig 4, an increase in the graft implantation angle resulted in a higher number of nanoparticles colliding with the surface, which increases the risk of stenosis. Surface No. 5 had the highest nanoparticle collision and risk of stenosis, followed by Surface No. 2. In geometries with implanted graft angles of 37.5 and 45˚, nanoparticles collided with Surface No. 1, indicating poor performance. The results showed a significant difference between the two inlet velocity conditions, indicating that the effect of pulsatile flow cannot be neglected. Although the overall trend was similar for the number of released nanoparticles, the number of colliding particles varied depending on the release conditions, with different NCP values.

**Table 3. NCP various surfaces for grafts with different implantation angles with permeation from wall and the application of all forces.**

| Surfaces | graft implant angle [˚] | | | | | | | | | |
|---|---|---|---|---|---|---|---|---|---|---|
| | 15 | | 22.5 | | 30 | | 37.5 | | 45 | |
| | CIV | PIV | CIV | PIV | CIV | PIV | CIV | PIV | CIV | PIV |
| Release of 100 nanoparticles in each time step (The total number of released nanoparticles is 3000) | | | | | | | | | | |
| 1 | 0 | 0 | 0 | 0 | 0 | 17 | 173 | 273 | 507 | 853 |
| 2 | 17 | 57 | 30 | 83 | 120 | 170 | 153 | 183 | 350 | 327 |
| 3 | 0 | 0 | 0 | 0 | 0 | 0 | 0 | 0 | 0 | 0 |
| 4 | 0 | 0 | 0 | 0 | 0 | 0 | 0 | 0 | 0 | 0 |
| 5 | 37 | 10 | 287 | 320 | 390 | 527 | 1103 | 633 | 2547 | 2267 |
| 6 | 0 | 0 | 0 | 0 | 0 | 0 | 0 | 0 | 0 | 17 |
| Release of 500 nanoparticles in each time step (The total number of released nanoparticles is 15000) | | | | | | | | | | |
| 1 | 0 | 0 | 0 | 0 | 0 | 27 | 177 | 967 | 406 | 1059 |
| 2 | 12 | 64 | 33 | 87 | 144 | 236 | 155 | 273 | 321 | 340 |
| 3 | 0 | 0 | 0 | 0 | 0 | 0 | 0 | 0 | 0 | 0 |
| 4 | 0 | 0 | 0 | 0 | 0 | 0 | 0 | 0 | 0 | 0 |
| 5 | 34 | 37 | 275 | 309 | 401 | 427 | 1037 | 584 | 2447 | 1753 |
| 6 | 0 | 0 | 0 | 0 | 0 | 0 | 0 | 0 | 24 | 12 |
| Release of 1000 nanoparticles in each time step (The total number of released nanoparticles is 30000) | | | | | | | | | | |
| 1 | 0 | 0 | 0 | 0 | 6 | 34 | 174 | 1055 | 589 | 1646 |
| 2 | 10 | 50 | 35 | 77 | 135 | 257 | 154 | 286 | 386 | 421 |
| 3 | 0 | 0 | 0 | 0 | 0 | 0 | 0 | 0 | 0 | 0 |
| 4 | 0 | 0 | 0 | 0 | 0 | 0 | 0 | 0 | 0 | 0 |
| 5 | 39 | 58 | 302 | 251 | 398 | 437 | 1093 | 942 | 2407 | 2447 |
| 6 | 0 | 0 | 0 | 0 | 0 | 0 | 0 | 0 | 30 | 21 |

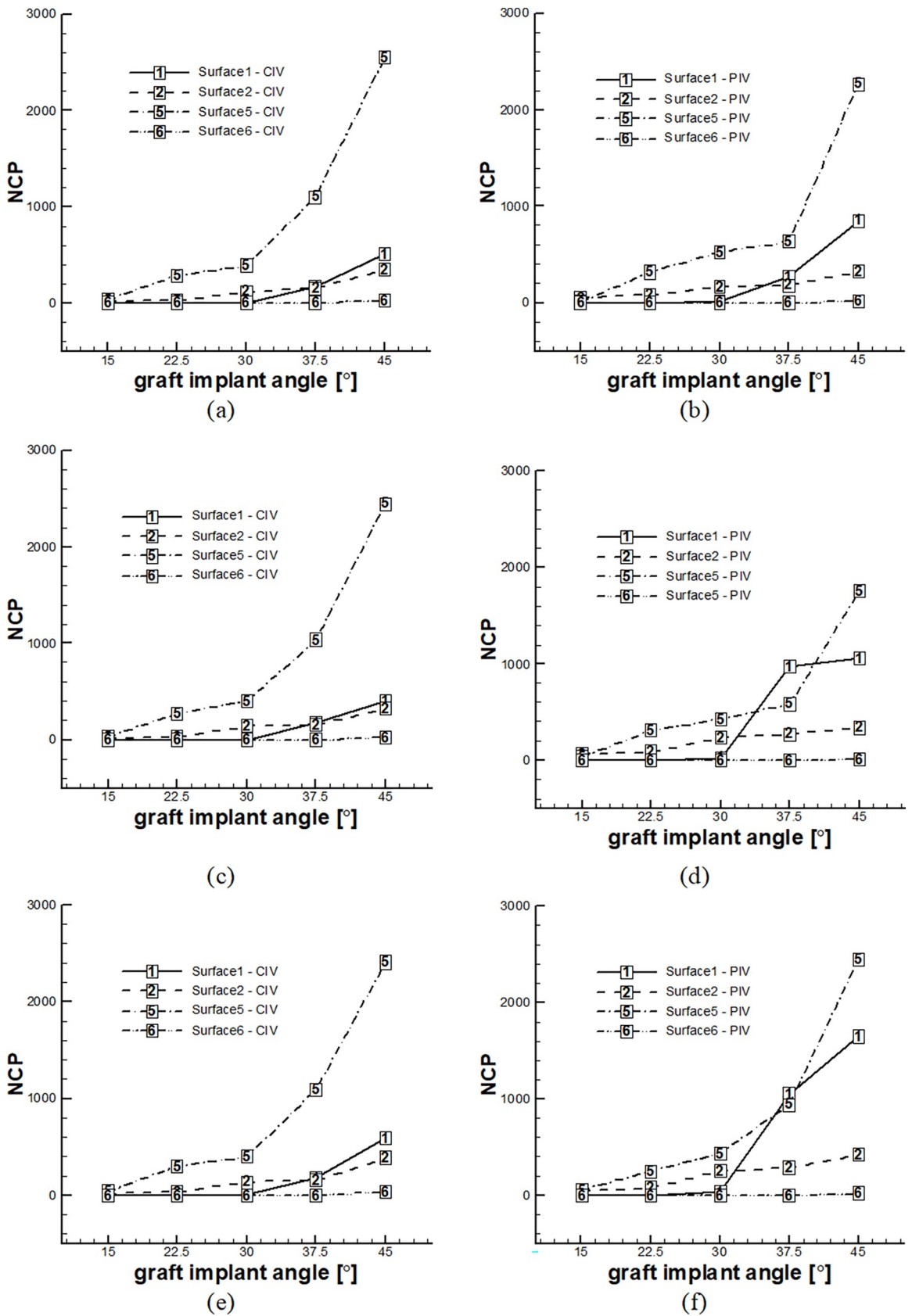

**Fig 4. NCP of various surfaces according to the graft's implantation angle and various number of injection of nanoparticles at the inlet at each time step with permeation from the wall and the application of all forces.** A) CIV and injection of 100 nanoparticle in each step time B) PIV and injection of 100 nanoparticle in each step time C) CIV and injection of 500 nanoparticle in each step time D) PIV and injection of 500 nanoparticle in each step time E) CIV and injection of 1000 nanoparticle in each step time F) PIV and injection of 1000 nanoparticle in each step time.

Table 4 and Fig 5 display the total number of NCPs values across various surfaces in all geometries, indicating that Surface No. 5 experienced the highest overall collisions, followed by Surface No. 1. Specifically, at implantation angles of 37.5˚ and 45˚, Surface No. 1 exhibited relatively high nanoparticle collisions, with Surface No. 2 ranking second overall after Surface No. 5. The study's findings indicate that the graft's outlet and inlet registered the highest number of colliding nanoparticles and showed the highest number of points with a heightened risk of stenosis. Furthermore, the results underscore the differences observed in various modes of nanoparticle release.

Table 4 and Fig 5 also provide information on the total number of NCPs in various geometries. According to the presented data, the graft implanted at an angle of 15˚ demonstrated the longest lifespan, as it exhibited the lowest number of trapped nanoparticles. A greater number of nanoparticles colliding with the surface increases the risk of stenosis, whereas a smaller number of colliding nanoparticles prolongs the duration of stenosis development. As the graft implantation angle increases, the lifetime of the graft decreases, which is apparent in the different number of released nanoparticles and the applied inlet velocity (constant and pulsatile). These findings also suggest differences between the CIV and PIV regimes.

## 4.4. Wall shear stress

In Fig 6A, the wall shear stress (WSS) is compared across various geometries. The higher the WSS, the fewer nanoparticles collide with the wall, resulting in less particle build-up due to the stronger shear flow near the wall. As shown in Fig 6A, decreasing the graft implantation angle increases the number of high shear stress surfaces along the graft. Additionally, the WSS on different surfaces decreases as the implanted graft angle increases. Therefore, the graft implanted at a 15˚ angle exhibited better performance in terms of nanoparticles colliding with surfaces and subsequent stenosis.

The study focused on flow behavior in cross-section A (see Fig 3), with Fig 6B presenting flow parameters for a graft implantation angle of 30˚, which also applied to other geometries. A secondary flow arose in the channel's center due to its curvature and the highest longitudinal

**Table 4. The total number of NCPs according various surfaces and geometries.**

| | | Summation of NCPs | | | | | | | | | | |
|---|---|---|---|---|---|---|---|---|---|---|---|---|
| | | Surfaces | | | | | | graft implant angle [˚] | | | | |
| | | 1 | 2 | 3 | 4 | 5 | 6 | 15 | 22.5 | 30 | 37.5 | 45 |
| Release of 100 nanoparticles in each time step | CIV | 680 | 670 | 0 | 0 | 4363 | 27 | 53 | 317 | 510 | 1430 | 3430 |
| | PIV | 1143 | 820 | 0 | 0 | 3757 | 17 | 67 | 403 | 713 | 1090 | 3463 |
| Release of 500 nanoparticles in each time step | CIV | 583 | 665 | 0 | 0 | 4195 | 24 | 46 | 308 | 545 | 1369 | 3199 |
| | PIV | 2053 | 1000 | 0 | 0 | 3110 | 12 | 101 | 396 | 690 | 1824 | 3165 |
| Release of 1000 nanoparticles in each time step | CIV | 769 | 720 | 0 | 0 | 4238 | 30 | 49 | 337 | 539 | 1421 | 3411 |
| | PIV | 2735 | 1091 | 0 | 0 | 4135 | 21 | 108 | 328 | 727 | 2283 | 4535 |

With permeation from the wall and with, two states of CIV and PIV and injection of 100, 500 and 1000 nanoparticles at inlet at each time step and applying all the forces.

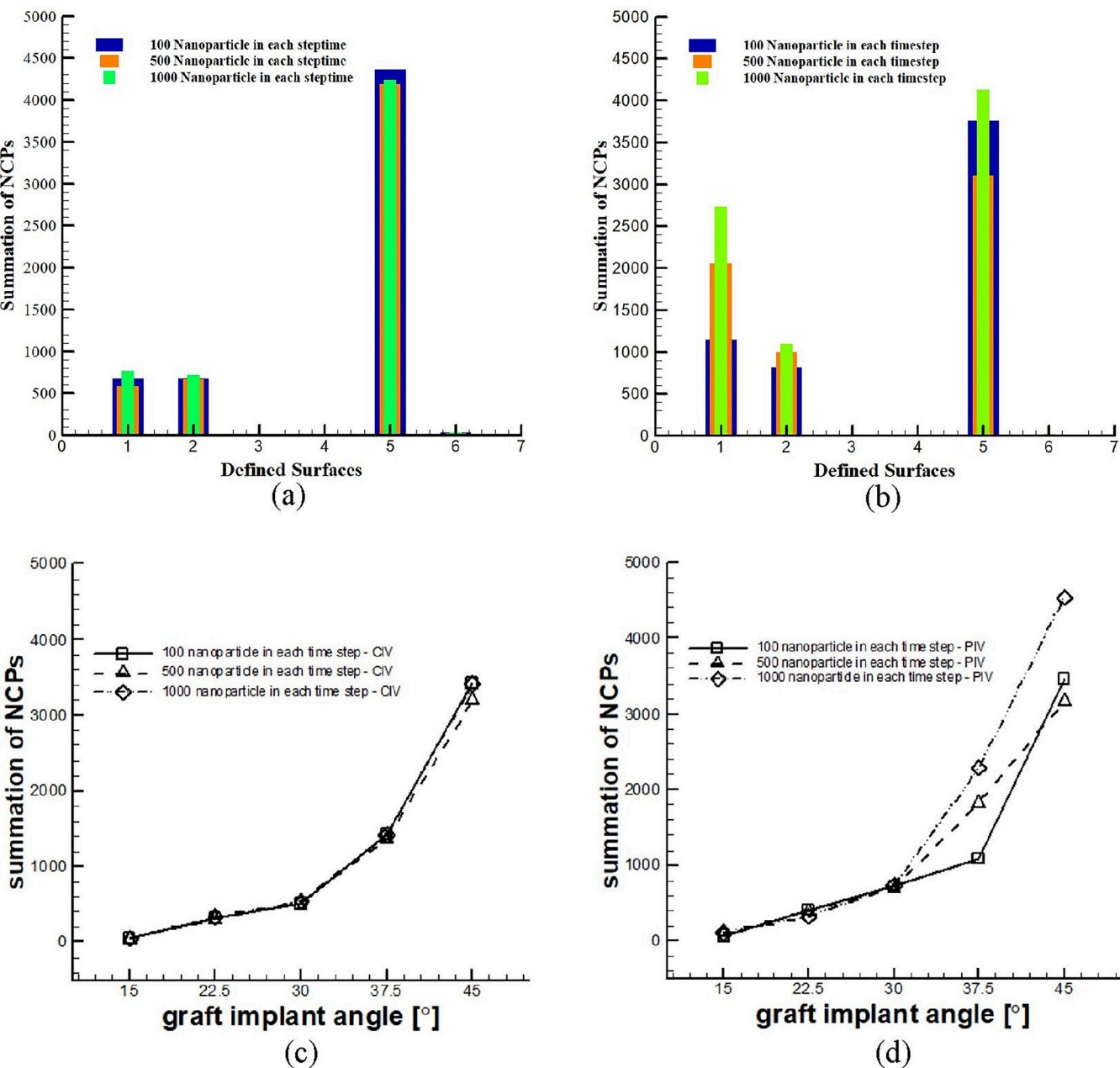

**Fig 5. The total number of NCPs according to various surfaces and geometries with permeation from the wall and with two states of CIV and PIV, with the injection of 100, 500, and 1000 nanoparticles at the inlet at each time step and applying all the forces.** a) CIV on various surfaces b) PIV on various surfaces c) CIV on various geometries d) PIV on various geometries.

velocity being in the cross-section. This caused a secondary rotational velocity field across the channel. Investigating the impact of these secondary velocity fields on nanoparticle movement could be a potential area for future research.

## 5.Conclusion

This study found that surfaces with more colliding nanoparticles are more likely to develop stenosis. Using one-way simulation of two-phase flow (blood and LDL nanoparticles), smaller

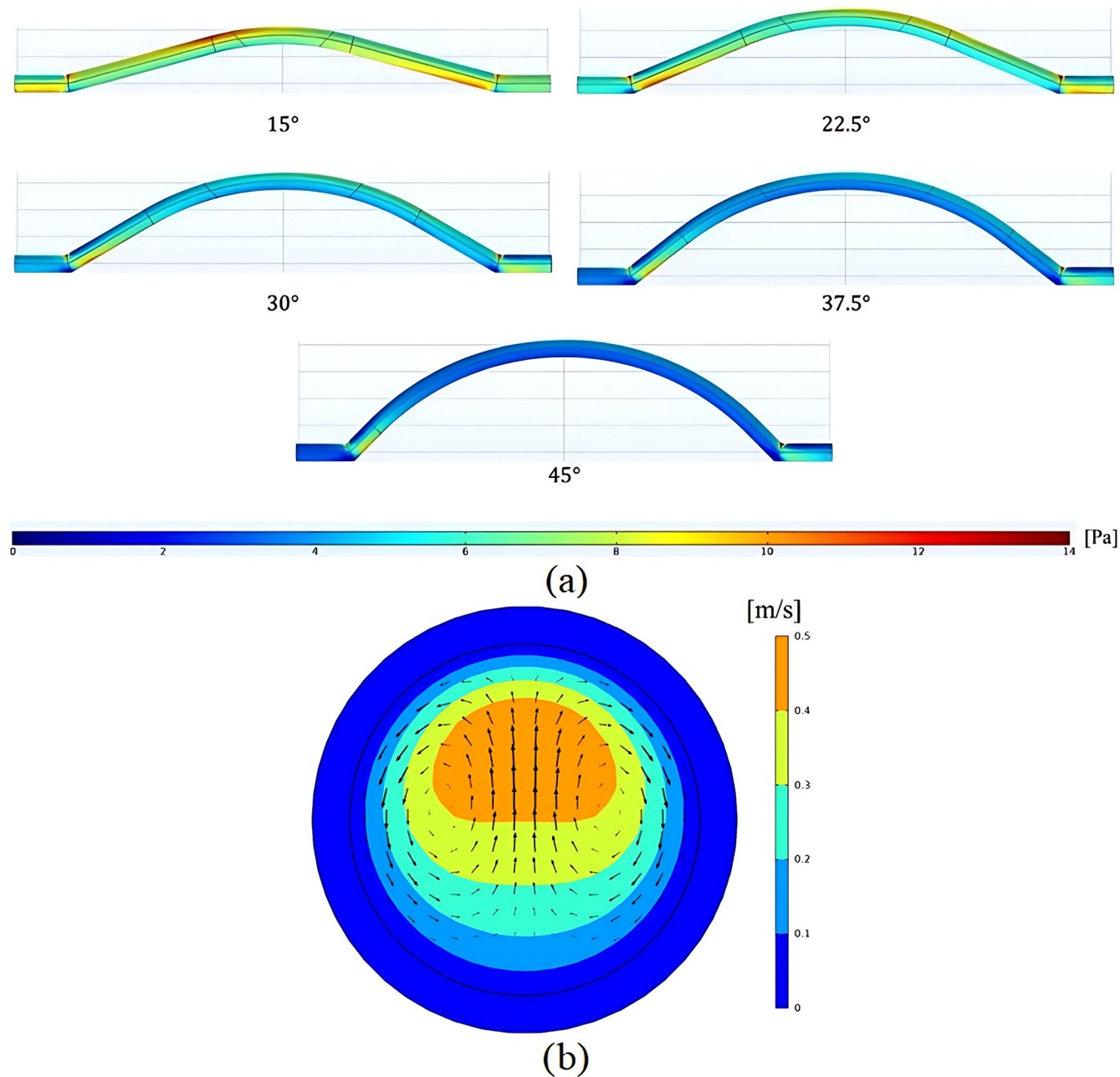

**Fig 6.** A) WSS in grafts with versus implantation angles B) Axial velocity contour and secondary flow velocity vectors occurred in section A (see Fig 3) in graft with 30° implantation angle.

implantation angles for grafts resulted in fewer nanoparticles colliding with the graft wall. The graft implanted at 15° showed the best performance, followed by the 22.5° angle. The graft inlet and outlet had the highest collision rate and therefore a higher probability of stenosis. Comparing permeation and non-permeation of walls, a permeable wall increased the NCP, indicating the effect CP of LDLs on nanoparticle collision and stenosis. The Saffman induced lift and wall-induced lift forces had no effect and can be ignored. No difference was found in the collision site of nanoparticles with different injection modes, but NCP values showed

dependence on the number of released nanoparticles. Constant and pulsatile inlet velocity modes showed differences in the number of trapped nanoparticles on surfaces. High-stress surfaces decreased with larger implantation angles, resulting in more nanoparticles colliding with the surfaces and intensifying stenosis.

The comparison of simulated geometries indicates that the graft implantation angles of 37.5˚ and 45˚ have higher numbers of trapped particles due to their additional surfaces area. Among the graft implantation angles of 15˚, 22.5˚, and 30˚, the 15˚ angle is preferred as it results in a lower number of colliding nanoparticles, leading to a decrease in stenosis and a longer graft lifetime. Generally, a smaller implantation angle, shorter graft length, and lower curvature decrease the risk of re-occlusion and graft failure. It's worth noting that only LDL nanoparticles were considered in this study, and the accuracy of the results could be improved by considering other blood components such as plasma, white and red blood cells and etc.

## Supporting information

**S1 File. Software raw files.**
(RAR)

**S1 Table. S1 Raw data of Fig 1B (pulsatile inlet velocity).**
(XLSX)

**S2 Table. S2 Raw data of Fig 2.**
(XLSX)

**S3 Table. S3 Raw data of Fig 4.**
(XLSX)

**S4 Table. S4 Raw data of Fig 5.**
(XLSX)

## Author Contributions

**Conceptualization:** Reza Karimian.

**Data curation:** Reza Karimian.

**Investigation:** Reza Karimian.

**Methodology:** Reza Karimian.

**Project administration:** Reza Karimian.

**Software:** Reza Karimian.

**Supervision:** Mohsen Saghafian, Ebrahim Shirani.

**Validation:** Reza Karimian.

**Writing – original draft:** Reza Karimian.

**Writing – review & editing:** Reza Karimian.

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
