## [Decision Letter · Decision Letter 0]

18 Oct 2023

PONE-D-23-27624Numerical investigation of LDL nanoparticle collision in coronary artery grafts with porous wall and different implantation angles and two state of inlet velocityPLOS ONE

Dear Dr. Karimian,

Thank you for submitting your manuscript to PLOS ONE. After careful consideration, we feel that it has merit but does not fully meet PLOS ONE’s publication criteria as it currently stands. Therefore, we invite you to submit a revised version of the manuscript that addresses the points raised during the review process.

In addition to the reviewers' comments, the authors are suggested to include justification for assuming flow as Newtonian as blood flow is mostly non-Newtonian 

We look forward to receiving your revised manuscript.

Kind regards,

Muhammad Shakaib, PhD

Academic Editor

PLOS ONE

4. PLOS requires an ORCID iD for the corresponding author in Editorial Manager on papers submitted after December 6th, 2016. Please ensure that you have an ORCID iD and that it is validated in Editorial Manager. To do this, go to ‘Update my Information’ (in the upper left-hand corner of the main menu), and click on the Fetch/Validate link next to the ORCID field. This will take you to the ORCID site and allow you to create a new iD or authenticate a pre-existing iD in Editorial Manager. Please see the following video for instructions on linking an ORCID iD to your Editorial Manager account: "" ext-link-type="uri" xlink:type="simple">https://www.youtube.com/watch?v=_xcclfuvtxQ"".

5. We note that Figure 6 in your submission contain copyrighted images. All PLOS content is published under the Creative Commons Attribution License (CC BY 4.0), which means that the manuscript, images, and Supporting Information files will be freely available online, and any third party is permitted to access, download, copy, distribute, and use these materials in any way, even commercially, with proper attribution. For more information, see our copyright guidelines: http://journals.plos.org/plosone/s/licenses-and-copyright.

a. You may seek permission from the original copyright holder of Figure 6 to publish the content specifically under the CC BY 4.0 license. 

Reviewers' comments:

Reviewer's Responses to Questions

**Comments to the Author**

1. Is the manuscript technically sound, and do the data support the conclusions?

Reviewer #1: Yes

Reviewer #2: Yes

2. Has the statistical analysis been performed appropriately and rigorously? 

Reviewer #1: Yes

Reviewer #2: No

3. Have the authors made all data underlying the findings in their manuscript fully available?

Reviewer #1: Yes

Reviewer #2: No

4. Is the manuscript presented in an intelligible fashion and written in standard English?

Reviewer #1: Yes

Reviewer #2: Yes

5. Review Comments to the Author

Reviewer #1: This is an interesting study that presents results of numerical simulations of LDL aggregation on the vessel walls, in various positions of coronary artery grafts.

The paper is properly organized. The addressed problem is properly explained, the simulation setup is described clearly. The mesh independence study was performed and the simulation results are validated against data from literature. The results are well presented and discussed.

There are some corrections that would improve the quality of the paper.

The abbreviation CP was used prior to being introduced in the main text of the manuscript.

Overview of the organization of the paper should be included at the end of the Introduction Section.

Page 4, Lines 90-92 – the value of outlet velocity from the artery is written to be identical for both considered cases …?

It is written that the software COMSOL multiphysics was used for the simulations, but some more details should be provided – that it was finite element method, which time step etc.

Besides the provided Figures, some quantitative data should be provided for the Validation section, per example the SD error or similar measurements.

Page 7, Lines 137 and 138 – This sentence should be rewritten to ensure better clarity.

Page 10, Line 220 – The reader is referred to a cross-section B shown in Fig. 3, however, in Fig. 3 only cross-section A is denoted. In the caption of Fig. 6, it is written that the considered cross-section is denoted with A, so I suppose this was a typo.

The paper provides an interesting and useful analysis of the effect of graft implantation angle on the aggregation of LDL particles within the walls. These results and further possible analyses using the proposed numerical approach can be useful for the improvements of clinical treatments. There are some corrections that are mentioned in this Review that would additionally improve the manuscript. I think that with these corrections this manuscript can be accepted for publication.

Reviewer #2: In this work, the authors provide a good numerical analysis of blood flow through coronary graft considering LDL particles. Generally, the approach is appreciated; however, some major concerns need to be addressed by the authors as detailed below:

1- In the introduction, no updated references are used. How can the authors claim that something done is new and contains the necessary innovation?

2- Obviously, pulsatile flow gives more realistic results because it is not practically constant. Why does the author assume a constant rate and then compare the results with the real state?

3- In pulsatile flow, the most important parameter is OSI, which is in fact the main criterion for evaluating the rate of disease progression. The author should plot the OSI values for different angles of graft and then do the necessary analysis on the optimal angle.

4- The porosity of a patient`s artery is different from that of a healthy person. One of the reasons for the progression of arteriosclerosis is this difference in porosity. Have the authors considered this?

5- In the simulation the length of artery entrance is taken to be 10 mm. Has the flow reached a state of development during length?

6- Is there any research that focuses on optimal angle of graft from another perspective? What is their optimal angle? Their results should be compared with the results of current work.

7- There are many grammatical errors in the text and the structure of some sentences is unclear to the readers. The entire text should be revised from this point of view.

8- The information in reference 8 is incomplete.

6. PLOS authors have the option to publish the peer review history of their article (what does this mean?). If published, this will include your full peer review and any attached files.

Reviewer #1: No

Reviewer #2: **Yes: **Safoora Karimi

---

## [Author Response · Author response to Decision Letter 0]

8 Feb 2024

Dear Editor,

We express our sincere gratitude to you and the reviewers for the precise review of our article and for providing insightful comments aimed at enhancing the quality of our work. We have carefully reviewed and addressed all the comments raised by the reviewers, implementing necessary corrections and highlighting these changes throughout the manuscript. Additionally, we have attached our detailed responses addressing the queries posed by the reviewers.

Answer to editor's comment:

In addition to the reviewers' comments, the authors are suggested to include justification for assuming flow as Newtonian as blood flow is mostly non-Newtonian.

In vessels with larger diameters, blood behavior tends to exhibit Newtonian characteristics. Coronary arteries, known for their relatively larger diameter within the human body, often showcase blood flow that aligns with Newtonian behavior. Considering this, we assume the blood flow in these vessels to be predominantly Newtonian. Additionally, in response to the editor's comment, we have incorporated references [14, 15] on page 5, line 121.

Answer to Reviewer 1:

 The abbreviation CP was used prior to being introduced in the main text of the manuscript.

The abbreviation CP and its definition were initially introduced in the Abstract. However, to ensure clarity for readers, we have included the corresponding abbreviation on page 3, lines 53 and 54 of the manuscript.

 Overview of the organization of the paper should be included at the end of the Introduction Section.

Following the esteemed reviewer's suggestion, we have included an overview of the paper's organization at the end of the Introduction. We think this addition provides a clearer roadmap of the paper's structure and content. The overview is now presented on page 4, between lines 96 to 98, as follows:

"In Section 2, the paper discusses the problem definition, followed by the presentation of geometry and equations. Section 3 covers the grid study and validation of the obtained results. The subsequent section, Section 4, presents and analyzes the results. Finally, Section 5 encapsulates the conclusions drawn from this study."

 Page 4, Lines 90-92 – the value of outlet velocity from the artery is written to be identical for both considered cases …?

On page 5, lines 133 to 134, the values of the outlet velocity from the vessel wall are considered to be the same in both cases.

 It is written that the software COMSOL Multiphysics was used for the simulations, but some more details should be provided – that it was finite element method, which time step etc.

Per the esteemed reviewer's valuable suggestion, amendments have been incorporated as follows, specifically on page 4, between lines 116 and 118:

"COMSOL Multiphysics, utilizing the finite element method, was employed for simulating the system dynamics. The simulations were conducted over three cardiac cycles."

 Besides the provided Figures, some quantitative data should be provided for the Validation section, per example the SD error or similar measurements.

During validation, the authors carefully compared the calculated results with those of the benchmark paper chosen for validation. The findings concluded that differences of less than 5% were observed in the maximum discrepancy cases. Nevertheless, in response to the valuable suggestions of the esteemed reviewer, specific amendments have been implemented, particularly on page 8, between lines 175 and 177 as below:

The comparison between the results of the paper chosen for validation and the simulations in the present study indicates differences of less than 5% in the cases with the maximum discrepancies.

 Page 7, Lines 137 and 138 – This sentence should be rewritten to ensure better clarity.

Thank you for your valuable input. We have revised the sentence on page 8, now found between lines 185 and 187, for improved clarity. The updated sentence reads as follows:

”Surfaces exhibiting higher collision rates are prone to developing stenosis. In this study, specific surfaces were defined, as illustrated in Fig 3.”

 Page 10, Line 220 – The reader is referred to a cross-section B shown in Fig. 3, however, in Fig. 3 only cross-section A is denoted. In the caption of Fig. 6, it is written that the considered cross-section is denoted with A, so I suppose this was a typo.

Thanks for the accuracy of the respected reviewer, as you correctly pointed out, it was a typo that was corrected on page 12, line 277.

Answer to reviewer 2:

 In the introduction, no updated references are used. How can the authors claim that something done is new and contains the necessary innovation?

As per the esteemed reviewer's suggestion, the introduction section has been thoroughly revised in the updated version of the paper. To address the concern regarding updated references, we have included the following papers within the revised introduction, precisely on pages 3 and 4, between lines 67 and 90: [5], [6], [7], [8]. These additions aim to reinforce the novelty and innovative aspects of the research conducted in the paper.

 Obviously, pulsatile flow gives more realistic results because it is not practically constant. Why does the author assume a constant rate and then compare the results with the real state?

Upon reviewing the existing literature on coronary artery simulations, it was observed that some previous researchers had indeed assumed constant blood flow in their studies. Acknowledging the reviewer's valid point, our study aimed to comprehensively investigate both scenarios: constant flow and pulsatile flow, with the intent to compare and demonstrate the discrepancies arising from non-pulsatile flow assumptions. It is evident from our findings that pulsatile flow exhibits higher accuracy and better aligns with real-world conditions, thereby supporting the claim that pulsatile flow is more representative of physiological reality.

 In pulsatile flow, the most important parameter is OSI, which is in fact the main criterion for evaluating the rate of disease progression. The author should plot the OSI values for different angles of graft and then do the necessary analysis on the optimal angle.

According to the OSI calculation formula which is as follows:

OSI=0.5*(1-|∫_0^T▒〖τ_w dT〗|/(∫_0^T▒|τ_w | dT))

The conclusion drawn is that, within the context of fluid flow in a channel, when the shear stress direction remains constant, the Oscillatory Shear Index (OSI) equals zero. Only at points of flow separation and reattachment does the shear stress direction change, resulting in a non-zero OSI value. Considering the specific geometry analyzed in this study, as flow separation does not occur at any point, OSI remains consistently zero across all locations. Consequently, this parameter is not utilized within the scope of this paper. The primary focus of this investigation lies in examining nanoparticle-wall collisions. Although OSI identifies potential locations for clogging due to flow separation, it does not serve the purpose of indicating surfaces affected by nanoparticle collisions.

 The porosity of a patient`s artery is different from that of a healthy person. One of the reasons for the progression of arteriosclerosis is this difference in porosity. Have the authors considered this?

The authors have referenced information from source [19] regarding porosity. However, it's important to note that there is generally limited available information specifically related to vascular porosity.

 In the simulation the length of artery entrance is taken to be 10 mm. Has the flow reached a state of development during length?

To apply the inlet velocity profile in our simulation, we selected the fully developed flow condition within the software, followed by the selection of velocity sub-branch. The selected conditions allowed us to simulate the flow as if it had reached a fully developed state from the very beginning.

 Is there any research that focuses on optimal angle of graft from another perspective? What is their optimal angle? Their results should be compared with the results of current work.

To the best of our knowledge, no specific optimal value for the graft implantation angle has been presented in the referenced literature. However, it is noteworthy that numerous researchers have commonly favored the lowest angle as the optimal choice for graft implantation such as Fan et al. (reference [2]).

 There are many grammatical errors in the text and the structure of some sentences is unclear to the readers. The entire text should be revised from this point of view.

Thank you for the valuable feedback regarding the grammatical errors and unclear sentence structures in the text. We have reviewed and revised the text accordingly to address these concerns.

 The information in reference 8 is incomplete.

The details from reference [8] have been revised, and the necessary corrections were made to page 13, specifically lines 333 to 334, (reference [9]).

---

## [Decision Letter · Decision Letter 1]

27 Feb 2024

Numerical investigation of LDL nanoparticle collision in coronary artery grafts with porous wall and different implantation angles and two state of inlet velocity

PONE-D-23-27624R1

Dear Dr. Karimian,

We’re pleased to inform you that your manuscript has been judged scientifically suitable for publication and will be formally accepted for publication once it meets all outstanding technical requirements.

Kind regards,

Muhammad Shakaib, PhD

Academic Editor

PLOS ONE

Additional Editor Comments (optional):

Reviewers' comments:

Reviewer's Responses to Questions

**Comments to the Author**

1. If the authors have adequately addressed your comments raised in a previous round of review and you feel that this manuscript is now acceptable for publication, you may indicate that here to bypass the “Comments to the Author” section, enter your conflict of interest statement in the “Confidential to Editor” section, and submit your "Accept" recommendation.

Reviewer #1: All comments have been addressed

Reviewer #2: All comments have been addressed

2. Is the manuscript technically sound, and do the data support the conclusions?

Reviewer #1: Yes

Reviewer #2: Yes

3. Has the statistical analysis been performed appropriately and rigorously? 

Reviewer #1: Yes

Reviewer #2: Yes

4. Have the authors made all data underlying the findings in their manuscript fully available?

Reviewer #1: Yes

Reviewer #2: Yes

5. Is the manuscript presented in an intelligible fashion and written in standard English?

Reviewer #1: Yes

Reviewer #2: Yes

6. Review Comments to the Author

Reviewer #1: The authors considered all the suggestions from the previous review and modified the manuscript accordingly. All the other changes in the revised manuscript improved it significantly. I think that this revised manuscript can be accepted for publication.

Reviewer #2: (No Response)

7. PLOS authors have the option to publish the peer review history of their article (what does this mean?). If published, this will include your full peer review and any attached files.

Reviewer #1: No

Reviewer #2: No

---

## [Editor Report · Acceptance letter]

1 Apr 2024

PONE-D-23-27624R1 

PLOS ONE

Dear Dr. Karimian, 

I'm pleased to inform you that your manuscript has been deemed suitable for publication in PLOS ONE. Congratulations! Your manuscript is now being handed over to our production team.

Kind regards, 

on behalf of

Dr. Muhammad Shakaib 

Academic Editor

PLOS ONE